# Whole-Heart Tissue Engineering and Cardiac Patches: Challenges and Promises

**DOI:** 10.3390/bioengineering10010106

**Published:** 2023-01-12

**Authors:** Aram Akbarzadeh, Soheila Sobhani, Alireza Soltani Khaboushan, Abdol-Mohammad Kajbafzadeh

**Affiliations:** 1Pediatric Urology and Regenerative Medicine Research Center, Gene, Cell and Tissue Institute, Children’s Medical Center, Pediatric Center of Excellence, Tehran University of Medical Sciences, Tehran 1419733151, Iran; 2Students’ Scientific Research Center, Tehran University of Medical Sciences, Tehran 1419733151, Iran

**Keywords:** cardiac tissue engineering, whole-heart regeneration, decellularized extracellular matrix, cardiac patch

## Abstract

Despite all the advances in preventing, diagnosing, and treating cardiovascular disorders, they still account for a significant part of mortality and morbidity worldwide. The advent of tissue engineering and regenerative medicine has provided novel therapeutic approaches for the treatment of various diseases. Tissue engineering relies on three pillars: scaffolds, stem cells, and growth factors. Gene and cell therapy methods have been introduced as primary approaches to cardiac tissue engineering. Although the application of gene and cell therapy has resulted in improved regeneration of damaged cardiac tissue, further studies are needed to resolve their limitations, enhance their effectiveness, and translate them into the clinical setting. Scaffolds from synthetic, natural, or decellularized sources have provided desirable characteristics for the repair of cardiac tissue. Decellularized scaffolds are widely studied in heart regeneration, either as cell-free constructs or cell-seeded platforms. The application of human- or animal-derived decellularized heart patches has promoted the regeneration of heart tissue through in vivo and in vitro studies. Due to the complexity of cardiac tissue engineering, there is still a long way to go before cardiac patches or decellularized whole-heart scaffolds can be routinely used in clinical practice. This paper aims to review the decellularized whole-heart scaffolds and cardiac patches utilized in the regeneration of damaged cardiac tissue. Moreover, various decellularization methods related to these scaffolds will be discussed.

## 1. Introduction

Despite a half-century of advances in preventive medicine, cardiovascular diseases (CVDs) are still recognized as a major global health concern among noncommunicable diseases, which yield a high rate of morbidity and mortality worldwide [1]. After an ischemic injury, the lost cardiomyocytes (CMs) are replaced by fibrotic scar tissue. Due to the limited capacity of the adult mammalian heart for regeneration, whole-heart transplantation is the only viable option for patients with end-stage heart disease [2].

Regenerative medicine is recognized as a promising field of translational research that decreases the need for donor organs. The field of cardiac tissue engineering has been a point of interest in the past few decades, mainly regarding the whole decellularized heart as bioscaffolds with further repopulation to a functioning heart that is ready for transplantation; however, there are still many challenges to creating in vitro fully functioning cardiac tissue [1]. Decellularized scaffolds are unique, complex, and natural sources that can be repopulated with cardiac-specific endothelial cells, fibroblasts (FB), and human pluripotent stem cell-derived CMs (hPSC-CMs) under the electrical and mechanical guidance of 3D bioreactors. Recent studies confirmed that developing off-the-shelf bioartificial tissues is a favorable choice for implantation, considering the material expenses that can help overcome the shortcomings of donor hearts [3].

In this review, we discuss regenerative medicine advances in heart tissue regeneration and the potential of off-the-shelf cell-free systems for repairing heart damage without the pitfalls of organ donors.

## 2. Cardiac Cells

The cardiac muscle is the core unit of the heart involved in force generation and guidance of electrical signals through the cardiac tissue, that is rich in CMs to induce rhythmic pump contractions. CMs constitute approximately 30% to 40% of the whole cell numbers in adult mammalian hearts and more than 80% of the whole heart volume, while the remained cells primarily include FBs and endothelial cells (ECs) with a smaller population of vascular smooth muscle cells (SMCs)/pericytes and hematopoietic-derived cells [4,5].

## 3. Cardiac Extracellular Matrix

The study by Johnson et al. on decellularized adult human cardiac extracellular matrix (cECM) showed that fibrillar collagens, especially collagen types I and V, compose more than 70% of human cECM. Approximately 20% of human cECM is basement membrane with the main component of collagen type IV and other proteins, including laminin, agrin, perlecan, and nidogen. Of ECM, 4% is structural ECM, mainly consisting of proteoglycans and fibrous glycoproteins [6,7].

## 4. Regenerative Medicine: Current Therapies for Cardiac Diseases

The prevalence of acute myocardial infarction (MI) is considered to be almost half a million annually, which also remains a leading cause of death worldwide despite advanced therapies. In the following section, to-date approaches for cardiac TE are highlighted.

### 4.1. Gene Therapy

Gene therapy is considered the first generation of cardiac regenerative medicine, which involves reaching therapeutic levels of a protein that can improve heart function via either repairing protein levels that are essential for the normal cardiac system or via the knockdown of proteins that can affect heart function. The DNA that encodes the target protein must have access to the cardiomyocyte’s nucleus by administering it in plasmid form (pDNA) or viral vectors such as adenovirus, adeno-associated virus (AAV), and lentivirus [8]. It appears that gene transfer efficiency is a foremost problem of gene therapy in many cardiovascular trials [9]. In the case of plasmid transfection, the efficiency of naked DNA’s uptake by cardiomyocytes remains too low, and only for the first couple of weeks after naked plasmid injection, measurable levels of gene expression are detectable, which might not have a meaningful therapeutic effect [10]. Regarding adenoviral vectors (which are the most efficient method for gene transfer), these vectors have raised safety concerns since they can stimulate strong immune and inflammatory responses, which can cause death. So, the dose of vector that can be injected is limited [11]. AAV’s immunogenicity would be expected to be easily controlled by pharmacological treatment at the time of vector administration, which is opposed to adenovirus. However, the permanent expression of the transgene leads to a limit in the use of these vectors [12]. With the current knowledge, there is no method for the transcriptional or posttranscriptional control of gene expression in vivo [13].

Another challenge for gene therapy is determining which gene should be modified. It was soon noticed that a single gene does not lead to a complete repair, except for enzymatic deficiencies in which the single nucleotide polymorphisms would be sufficient in a catalytic effect. Soon the new clinical trials were primed for cardiovascular gene combinations. In 2019, a clinical trial (INXN-4001, NCT03409627) by infusion of a plasmid expressing S100A1, SDF1α, and VEGF-A in patients with an implanted left ventricular assist device was performed, and the efficiency of plasmid gene transfer and the efficacy of SDF1α seems unsatisfactory [14]. Moreover, it has become evident that an appropriate source of cells with a regenerative capacity is required for gene therapies to effectively express the required gene that was not studied in injured hearts. The first cardiovascular gene therapy in humans was performed in 1990 by Nabel and her team by reporting the possibility of direct intra-arterial gene transfer through endovascular catheter techniques [15]. After 30 years, almost 3000 clinical trials have been approved, conducted, or begun in different fields of medicine (http://www.abedia.com/wiley/, accessed on 20 September 2019); however, there is not even a single gene therapy that has a significant impact in the cardiovascular arena [16]. These mentioned recognitions led to the conclusion that gene therapy might not be the best option to solve the deficiencies following heart failure or acute MI [3].

### 4.2. Cell Therapy

The two complementary aims of regenerative cardiac cell therapy are the replacement of injured myocardium with contractile CMs and paracrine modification of the repair process in each contractile and noncontractile cell, including fibrosis, apoptosis, inflammation, and angiogenesis [17,18,19]. Cells can be delivered by intravenous, intracoronary, intramyocardial, and intrapericardial routes. Current delivery methods used in clinical trials have faced many caveats; for example, more than 90% of cells wash out almost immediately after the intramyocardial and intracoronary injection. Furthermore, a large number of cells migrate to the spleen or liver during the intracoronary injection [20]. The dose of the injected cells has an important role in improving myocardial function. By repeating injection cells, sufficient cells can deliver to the infarcted myocardium. Different policies involving the different combinations of dosing cells, delivery methods, and time intervals were suggested regarding repeated cell injection [21]. Yao et al. reported that, three months after the first transfer in patients with large MI, repeated bone marrow mononuclear cell infusion through the IC route could improve cardiac function [22]. Hsiao et al. showed a reduction of myocardial damage in patients by combining IC and IV administration [23].

Intramyocardial injection of stem cells increases the risk of secondary infections and injuries as it requires a thoracoscopic or open-chest surgery. The risk of proarrhythmic events also increases through reentry, automaticity, and triggered activity [24]. More than 200 trials have been performed over two decades for heart disease, and although the safety and feasibility of several different cell types were confirmed, they have so far failed to demonstrate clinical efficacy in the heart disease area [25]. Pioneer studies discussed using non-cardiomyogenic cell groups, especially bone marrow-derived mononuclear cells (BM-MNCs) or MSCs, which induce endogenous regenerative processes [18,26] and other cells, such as skeletal myoblasts adipose-derived regenerative cells (ADRCs) and hematopoietic stem cells (CD34+/CD133+) [27]. The second group of cells was used for cell therapies such as c-kit+ (CD117+) CSCs and cardiosphere-derived cells (CDCs) isolated from cardiac tissue. Clear paracrine signaling and multilineage trans differentiation were reported for CSCs in vitro [28]. CDCs formed self-adherent clusters composed of CSCs and supporting cells in vitro [29]. Allogeneic cells isolated from a young, healthy donor were preferable to old ones [30]. Moreover, in comparison with elderly patients, younger patients react better to stem cell therapy [31]. 

Pluripotent cells, such as embryonic stem cells, induced pluripotent stem cells placed in the second group, and lineage-directed stem cells, can be modified [32]. These cell sources with higher regenerative capacity need more reformed isolation and in vitro amplification processes, such as cardiac-derived progenitor cells and hPSC-CMs [18]. However, more studies are required to confirm the clinical outcome of these sources. The vast knowledge of pluripotent stem cell (PSC) technology combined with the advances in cell biology led to the ability to obtain any subgroups of CM, including pacemaker, atrial, or ventricular [33]. CM maturation is controlled by many factors, including electrical stimulation, mechanical forces, remodeling of ECM, and biochemical factors gradients. Multiple bioengineering mechanisms are proposed to promote the maturation of hPSC-CM [34]. 

### 4.3. Fabrication of Biomimetic Cardiac Tissues

Multiple innovative approaches are proposed to induce maturation in hPSC-CMs. In the following parts, we describe the most promising technologies adopted to date to grow biomimetic cardiac tissue in a laboratory [1].

#### 4.3.1. Hydrogels

Due to their biocompatibility and adjustable chemical and physical properties, hydrogels are considered one of the most suitable polymers for cardiac tissue engineering. Some advantages of using polymers include the possibility of containing cell-binding sites such as ECM proteins, RGD integrin-binding domains, high water content, scalability, and efficient exchange of metabolites. For engineering cardiac tissues, natural hydrogels [35] are ideal for supporting cell proliferation and subsequent maturation; however, hybrid biomaterials or functionalized synthetic polymers can be a proper alternative to natural ones [36]. For mimicking the architectural features of native tissues, scaffold porosity and microarchitecture should be controlled to achieve a functional tissue. The interconnected porous structure can be generated through solvent casting/particulate leaching, freeze-drying, gas foaming, and the unidirectional freezing technique [37]. Although hydrogels have numerous capabilities and advantages in cardiac tissue engineering, some limitations would become evident. A major limitation of these hydrogels remains their inability to integrate organized vasculature within engineered tissue. Another challenge is that cells often self-organize within hydrogels in ways different from those found in native tissues. Moreover, cell migration and proliferation can be affected by the mismatching between the pore size of the conventional hydrogel and the dimensions of cells [38]. Poor cell adhesion due to the inherent cell repellency of most hydrogels, such as poly (ethylene glycol) or zwitterionic hydrogels, can limit the use of hydrogels as functional engineered tissues [39]. The lower modulus of macroporous hydrogels, in comparison with natural tissues, can have an impact on CM maturation due to their inherent softness and high porosity [40]. The lack of biological complexity of the native cardiac ECM consists of proteoglycans, glycoproteins, fibrous protein growth factors, and other small molecules that are essential for adjusting cell−ECM crosstalk. Despite being a promising material, its potential use for cardiac tissue engineering still needs to be evaluated. 

#### 4.3.2. Microfabrication

Microfabrication allows the designed synthetic 3D cell niches to mimic the natural cell microenvironment. Surface patterns can be created by various techniques, such as lithography techniques (or microprinting), which involve photolithography and soft-lithography. Through photolithography, defined topographies for anisotropic tissue organization can be obtained by transferring the geometrical pattern from a photomask to a light-sensitive chemical on the substrate under the light. By means of soft lithography techniques, we can create tissue vasculature by designing microchannels and complex microfluidic networks using elastomeric stamps, molds, and photomasks [41]. However, producing such a platform encounters obstacles, such as the need for specialized equipment and facilities, and it can be time-consuming and costly. This fact led us to use microfabrication to pattern material substrates in a simpler system. One of the initial studies showed CM alignment was improved during the cardiac cycle when they were placed in channels by using polydimethylsiloxane (PDMS) thin films comprising an array of alternating 20-µm-wide lines [42]. Patterned culture chambers can help the long-term maturation of cells by supporting hydrogel-based cardiac tissue formation (~3–4 mm length) [43]. For online electrophysiological recording and electromechanical pacing, conductive biomaterials with built-in nanoelectronics (~20 mm × 5 mm) were applied [44]. By trying the simple soft lithography technique, cardiac tissues with perfusable and highly branched endothelialized channels as the vascular network can be assembled, while in the case of simple fabrication methods such as hydrogels, this is not able to be performed [45].

#### 4.3.3. Electrospinning

The highly aligned cellular and nanoscale ECM organization of native myocardium can be mimicked by nano-micron scale fibrous scaffolds [46]. Electrospinning is a versatile method to produce ECM-resembled scaffolds as knitted, woven, or braided fibrous networks, which can be fabricated from synthetic and naturally derived materials [47]. Promising results were reported regarding cardiac patch fabrication from electrospun nanofibrous scaffolds. One study showed poly(ε-caprolactone) (PCL) electrospun scaffolds (~2 cm × 2.5 cm × 229 nm) developed through solution electrospinning induced cell alignment in anisotropic electrospun fibers [48]. Another promising study proved that the CM alignment was promoted in a polyester scaffold (25 mm diameter) [49]. Moreover, suitable electrical stimuli to the cells are gained by coating the scaffold with piezoelectric microfibers. The fabricated human left ventricle (truncated ellipsoid; 4.5 mm diameter, 9 mm height) promoted the alignment of neonatal rat ventricular myocytes and cardiomyocytes derived from the human-induced pluripotent stem cells. The ejection fractions and contractile work were ~50–250 and ~104–108 smaller than the physiologic values rodent and human ventricles, respectively [50]. Melt electrospinning is another attractive methodology that allows more control and reproducibility over scaffold fabrication in comparison with solution electrospinning. Although the application of melt electrospun scaffolds for engineering is improper for cell encapsulation purposes, foldable PCL melt electrospun hexagonal-shaped scaffolds (8 mm diameter, 300 µm thick) promoted the CM alignment and maturation [51]. Some practical limitations in using electrospinning are the possible toxic effect of remaining chemical ingredients during the electrospinning postprocessing [22], an insufficient mechanical force for load-bearing aims [23], poor cell infiltration and migration of the cells [22,24,52], and the insufficient biochemical complexity of the cardiac ECM [53].

#### 4.3.4. Bioprinting

Bioprinting is an emerging biofabrication method for tissue-engineered construct fabrication through the precisely layer-by-layer deposition of biomaterials, biochemicals, and living cells [54]. Bioprinting technologies include laser-assisted printing [55], inkjet printing [56], and extrusion-based printing [57]. For printing, synthetic materials were the first in line, but now naturally derived materials, such as ECM hydrogels derived from decellularized tissue, capture attention. In particular, for patch-like construct fabrication, grid-type structures in the form of rectangles (~5 mm × 5 mm × 1 mm) [58] or circles (10 mm diameter, 0.6 mm thick) [59] are successfully generated from cardiac-specific ECM-derived bioinks by using extrusion-based 3D bioprinting. “FRESH” (i.e., Freeform Reversible Embedding of Suspended Hydrogels) is a new technology in which soft materials can be printed with low- or no-yield stress through printing bioink in a thermally reversible supportive hydrogel [60]. By using this technology, researchers designed the beating ventricle (truncated ellipsoid; 5.7 mm diameter, 8 mm height) and an acellular neonatal-sized heart analog (37 mm diameter, 55 mm height) [61]. In other research, the FRESH method can help print a heart-shaped model (14 mm diameter, 20 mm height) by applying a hydrogel derived from decellularized patient’s omentum as a bioink when combined with the patient’s own cells (CMs and endothelial cells) [62]. Consequently, the feasibility of fabricating thick, vascularized, and perfusable cardiac patches that match the patient’s immune system were examined. In the near future, by using personalized hydrogel, whole hearts with their major blood vessels can be printed. Another example of creating tissues with inherent perfusable vasculature is the producing vascularized cardiac tissue through direct extrusion-based bioprinting in which a microfluidic device was used to seed CM and perfuse the endothelialized grid-type structure (~5.5 mm × 3.5 mm × 0.75 mm) [25,26]. Stereolithography offers high printing speed with a high spatial resolution that uses a light projector to solidify the bioink layer-by-layer, leading to the formation of scruffy vascular networks to perfuse an alveolar sac model [63]. It is important to consider that the time-consuming process of printing with a high level of detail requires a slow printing velocity, affecting cell viability. On the other hand, it seems 3D bioprinting is better suited to small-scale printing tissues because organs like the heart and kidney have complicated structures and consist of different types of cells. In addition, biochemical characteristics and mechanical properties tremendously differ from one tissue to another and even within one tissue, e.g., renal cortex versus renal medulla [18].

#### 4.3.5. Decellularized Bioscaffolds

Decellularization methodology is used to produce acellular scaffolds with the naïve ECM structure. Here, several decellularization protocols are summarized for human organs and tissues, including cardiac tissue.

## 5. Decellularization and Recellularization Methods

### 5.1. Decellularization Agents

The selection of the most proper decellularization agent depends on many variables, including but not limited to the lipid content, cellularity, thickness, and density of tissues [64]. The objective of decellularization is to minimize the undesirable effects of solvent agents on ECM composition. Decellularization methods are divided mainly into three categories: biological (enzymes), chemical (ionic and non-ionic detergents), and physical (high hydrostatic pressure, freeze-thawing cycles, supercritical CO_2_) [65,66].

### 5.2. Techniques for Applying Decellularization Agents

The simple decellularization methods, including agitation, remove cellular components from tissues while preserving the original ultrastructure of the tissue (used for simply structured matrices, including heart valves) [67]. Perfusion-based decellularization protocols are proposed in solid organs for the most efficient removal of residual DNA and cells [68].

### 5.3. Evaluation of Decellularized ECM

One of the most significant concerns in decellularization is the mechanical and material preservation of the remaining ECM scaffold. Previous studies described that sodium dodecyl sulfate (SDS) and Triton X-100, two of the most used detergents, might disrupt collagen in certain tissues and reduce the mechanical strength of the tissue. However, some evidence reported no harm to the ultrastructure and mechanical properties in some tissues, including tendons [69]. Studies have revealed that some degrees of glycosaminoglycan (GAG) removal is confirmed for most detergents [70]. Figure 1 describes the impact of the decellularization and the recellularization process on cardiac ECM (cECM).

### 5.4. Recellularization and Cell Sources

Recellularization is the process of repopulating organ-specific cell types or stem cells into the acellular ECM scaffolds by the use of bioreactors. There are two main techniques for whole heart recellularization: intramyocardial injections and vascular networks [71].

Mechanically and electrically connected CMs construct the heart muscle, which is surrounded by FBs, heart resident immune cells, ECs, SMCs, and pericytes [72]. A three-dimensional (3D) construct is made by different types of cell sources, including CMs, CMs derived from induced pluripotent stem cells (iPSC-CM), CMs derived from embryonic stem cells (ESC-CM), EC, human umbilical vein endothelial cells (HUVEC), FBs, human cardiac fibroblasts (hcFB), and SMC and mesenchymal stromal cells (MSC) [73].

### 5.5. Bioreactors for Recellularization

However, the human or animal body serves as a natural bioreactor; a successful whole organ recellularization is dependent on the specific environment that mimics in vivo conditions of a specific organ. For example, organ-specific shear stress is required for the complete reendothelialization of the vascular tree [74]. Otherwise, the sudden exposure to blood flow after static cultivation would cause thrombotic complications and cellular rupture [75]. The real-time control of specific variables is vital, particularly during the long-term culture. These parameters include electrolyte levels, pH, pO2, pCO2, and perfusion factors [74].

### 5.6. Immunogenicity and Its Implications

The host response to the recellularized ECM scaffold might vary dependent on multiple factors, which can be either destructive or constructive. The constructive effect occurs when the scaffold is decellularized correctly, free of bacterial or fungal contaminations or endotoxins, and grafted into the healthy surrounding tissue. Successful decellularization is defined as a residual DNA concentration of less than 50 ng of dsDNA per mg of ECM with a fragment size of less than 200 bp [76]. On the other side, residual cellular material showed a destructive effect on the ECM by generating an immune response, which is classified into innate and adaptive responses [77]. Mast cells, dendritic cells, basophils, eosinophils, natural killer cells, neutrophils, and macrophages all can create the innate immune system, and the innate immune response can activate the adaptive immune system, which consists of memory cells [78].

## 6. Whole-Heart Engineering

The upcoming sections will review results for the decellularization and recellularization of whole mammalian hearts.

### 6.1. Development of Decellularization Protocols in Rodent Models

In 2008, the Ott lab pioneered the first technique to engineer a whole bioartificial heart when they recellularized neonatal rat CMs and rat ECs into the decellularized rat heart ECM scaffold, which resulted in almost 25% of neonatal heart function [79]. Following the pioneering work of Ott et al., multiple groups have applied decellularization technology to rat, mice, pig, and even human hearts [80,81,82].

Ng et al. reported that decellularized FVB/N mice hearts can differentiate stem/progenitor cells into cardiac lineage after the recellularization process. It was approved by expression of cardiac markers such ascTnT, Nkx-2.5, Myl2, Myl7, and Myh6 upon differentiation, but there is no beating observed in the recellularized scaffold after being implanted subcutaneously in SCID mice [81].

Akhyari et al. compared four protocols for whole-heart decellularization of LEW/Crl rats, which confirmed that none of the analyzed protocols produced a biological matrix entirely free of donor cell material and a scaffold with preserved ECM components [82]. Witzenburg et al. assessed decellularization’s effect on rats’ right ventricles’ mechanical characteristics. They confirmed that decellularized tissue introduces a valuable model for the native tissue ECM [83]. Crawford et al. showed the feasibility of cryopreserving rat hearts at −80 °C with 10% DMSO in PBS for one year and then recellularizing with canine endothelial cells. This study confirmed the successful use of cross-species cells and scaffolds [84].

To achieve a transplantable heart with the ability of further vascular anastomosis, the endothelialization of the vessel lumen within the cardiac scaffold is necessary to prevent thrombosis. In the first report of the re-endothelialization of whole decellularized hearts, Robertson et al. attempted to re-endothelialize whole decellularized hearts through both arterial and venous beds and cavities. Scaffold vessel re-endothelialization with rat aortic endothelial cells (RAECs) was increased in the combination of brachiocephalic artery (BA) and inferior vena cava (IVC) delivery strategy in comparison with single-route strategies [71]. Lu et al. reported the differentiation of human iPSC-derived cardiac progenitor cells to cardiac myocytes, smooth muscle cells, and ECs when seeded on whole decellularized C57BL6/ J mice hearts. The proliferation of cardiac myocytes in the presence of cECM can be stimulated for more time than the culturing of these cells in three-dimensional (3D) environments without ECM. Electrical activities, contractions, and mechanical forces were observed in the regenerated heart [85].

Tao et al. showed neonatal rat cardiac cells were positive for cardiac markers (α-actinin, cTnI, connexin 43), endothelial marker (vWF), and proliferation marker (ki67) after seeding on decellularized whole Sprague–Dawley rat heart. Within two days after repopulation, the heart began beating. This model only accounts for about 36% of the mass of an adult rat heart [86]. Following the seeded neonatal cardiac cells on cECM successfully differentiated into CMs, the researcher started using a neonatal cECM to promote the differentiation and maturation of neonatal cardiac cells.

When neonatal murine cardiomyocytes were seeded on decellularized neonatal mouse hearts, Garry et al. reported that cells were positive for both NKX2.5 and α-actinin, which is expressed by cardiac progenitor cells and differentiated CMs, respectively. Even after these cells undergo prolonged culture continue to express cardiomyocyte markers. So, neonatal matrices have enormous potential to be a novel construct for repopulation [87]. Whole heart decellularization and recellularization are performed through four-flow cannulation of the superior vena cava (SV), pulmonary vein (PV), ascending aorta (AA), and pulmonary artery (PA).

Four-flow cannulation preserves whole heart conformity, enabling ventricular pacing via the pulmonary vein. Following the recellularization process, around the vascular structures, HEK293 cells were seen, and seeded primary hcFBs were able to migrate and attach to the scaffolds. Growth of cell patches was observed macroscopically when the h-iPS-CPCs were perfused to the scaffold. For cardiovascular drug accuracy, 4-Flow cannulated rat hearts can use as the fundamental humanized organ model [88].

Numerous studies have been worked on decreasing detergent concentration and detergent exposure time to ensure that the ECM is well-preserved. In two studies, the detergent exposure time was decreased by using an electric field in the decellularization protocol [89] and using a pulsatile perfusion system since the scaffolds developed by the pulsatile perfusion showed significantly lower residual DNA content in comparison with the other scaffold developed with non-pulsatile perfusion [90]. In another study, Dal Sasso et al. used protease inhibition, antioxidation, and excitation to reduce the concentration and incubation time with cytotoxic detergents as well [91].

Alexanian et al. seeded the induced cardiac progenitor cells (iCPCs) fabricated through reprogramming adult mouse FBs, on the decellularized whole C57BL/6J mice heart to examine the safety and functionality of these cells. After three weeks, cells began migrating, colonizing, and finally differentiating in a scaffold which was demonstrated by detecting cardiac actin-positive CMs, SMA-positive smooth muscle cells, and CD31+. Electrically functional cardiomyocyte clusters have appeared in the scaffold under field stimulation [92]. The whole decellularized rabbit heart was reendothelialized by human iPSC-derived ECs and recellularized by hiPSCs-derived cardiac cells (CCs). The heterogeneous group of cardiac cells containing aligned cardiac troponin T-positive cells recovered LV wall thickness after 60 days. Results revealed maintaining vessel patency after transplanting this heart to the femoral artery bed of a pig and starting perfusion. Recellularized hearts exhibited visible beating [93].

Table 1 summarizes the to-date efforts on rodent whole-heart decellularization.

In general, the same concentrations of detergents and solutions used for murine hearts have been used for porcine hearts; however, solution volumes and exposure time have been increased [80].

### 6.2. Development of Decellularization Protocols in Human-Sized Models

Wainwright et al. introduced the first report of porcine whole heart decellularization in 2010, which recellularized lyophilized cECM sheets with 500,000 cells/cm^2^ of white leghorn chicken embryonic CMs [95].

Weymann et al. introduced an organ perfusion decellularization apparatus for the whole-heart TE of porcine hearts consisting of a pressure transducer that can control a roller pump by a computer system to maintain the perfusion pressure on 100 mmHg continuously, perform an air trap for a bubble-free perfusion, and use a heat exchanger to constantly keep the perfusate warm at 37 °C for the whole-heart TE of porcine hearts [96].

Weymann et al., for the first time, developed a tissue-engineered porcine human-sized whole heart with intrinsic electrical activity in the myocytes after recellularization [97]. In the first report of the heterotopic transplantation of a tissue-engineered heart, Kitahara et al. attempted to transplant a decellularized whole porcine heart and recellularized the whole heart with mesenchymal stem cells into pigs under systemic anticoagulation treatment with heparin. On day three, the hearts from both groups were harvested. Although short-term coronary artery perfusion in the transplanted scaffolds was observed by angiography, porcine MSCs were not found in the vessel lumen, and coronary thrombosis was detected in both groups [98]. Lee et al. showed that the decellularization of the whole porcine heart through inverting the heart offers a patent coronary vascular architecture with a higher coronary perfusion efficiency, leading to the removal of more native cells from ECM, improved ECM preservation, and, finally, the retention of its native macro- and microstructure [99].

Hodgson et al. used an automated pressure-controlled bioreactor for the decellularization of the porcine heart, which showed complete preservation of collagen and GAGs content and vascular network [100]. Akhyari et al. reported their results as a call for the reevaluation and improvement of the feasible decellularization methods. His team showed that, after whole-heart decellularization, the total protein content in the epicardial adipose tissue (EAT) as a regulator of cardiac anatomy and function was strongly reduced and large amounts of lipids were detected in EAT, indicated by lipid staining. However, there is no donor material in other regions of the heart; therefore, the perfusion decellularization of human-sized hearts shows inconsistent results regarding the different regions of the heart, which means all the main regions of the heart have to be evaluated separately. Therefore, in this paper, they studied five different parts of the heart individually. Incubation cardiac FBs with decellularized EAT showed a significantly diminished viability versus when incubating with native EAT or an unconditioned culture medium. Overall, the incomplete removal of donor material, residual detergents, and the removal of the protein during decellularization can negatively affect cell viability after the recellularization [101].

As we mentioned before, in 2019, our team decellularized whole ovine hearts through coronary perfusion. Later, different parts of the heart, such as the auricle, aortic valve, left and right ventricular myocardia, and chordae tendineae, were examined separately due to having a variable composition of ECM. The collagen and GAGs contents in chordae tendineae were decreased. The microvascular angiography indicated that the natural 3D architecture of the coronary tree was preserved. Subcutaneously implanted dECM into the omentum of Sprague–Dawley (host) rat showed, after two months, good vascularization and repopulation of the graft, indicated by the existence of CD31+, CD34+, and SMCs [102].

Table 2 summarizes porcine/ovine whole-heart decellularization techniques to date.

### 6.3. Development of Decellularization Protocols for Human Tissue

Sanchez et al. seeded human cardiac progenitor cells (hCPC), bone-marrow mesenchymal cells (hBMSCs), HUVECs, and H9c1 and HL-1 CMs into a decellularized human left ventricle. Results showed that the endocardium and vasculature were covered by HUVECs, and differentiated CMs organized properly into nascent muscle bundles [105]. Guyette et al. decellularized 73 human hearts. Cadaveric and decellularized human myocardia were subcutaneously implanted in Sprague–Dawley rats and harvested after two weeks. CD68+ mononuclear cells in two groups were detected. M2 macrophages appear in greater quantities in the decellularized human myocardium in comparison with a cadaveric human in proinflammatory response. The whole heart reseeded them with ~500 million human BJ fibroblast RNA-induced pluripotent stem cells (BJ RiPS)-derived CMs. On day 7, electrical stimulation was applied to the engineered tissue, and after 14 days, visible contractions were detectable [106].

In a novel method for human whole-heart decellularization, a pressurized pouch was used to keep the aortic valve closed during the perfusion of the detergents into the myocardium, thus enhancing the myocardial blood flow, resulting in improved decellularized tissue [67]. Table 3 describes human whole-heart decellularization methods to date.

## 7. Hurdles Should Be Addressed for Whole-Heart Engineering

More than four billion CMs and ECs comprise most of an adult human heart, making it very difficult to isolate and expand in large quantities [108]. The risk of teratoma formation increases when using any pluripotent progenitor cells [109]. The use of autologous ECs (iPS-derived ECs, blood-derived endothelial cells, or bone marrow-derived endothelial progenitor cells) is crucial to prevent thrombosis. It can solve the α-gal immunogenicity problems, making it suitable for clinical use [71]. Two of the most apparent issues in the heart, arrhythmogenesis, and thrombogenesis, are solved by the uniform recellularization of both the vascular tree and the parenchyma [110].

The other challenge for whole heart engineering is that despite the excellent effect of bioreactors, they cannot offer in vivo conditions for the whole heart. Bioreactors lack the complex interactions needed for tissues in the human body [110].

One of the significant challenges is re-establishing the oxygen and carbon substrate delivery to the engineered heart tissue within the first few hours after the transplantation [2]. The risk of potential thrombosis increases because of incomplete endothelialization and insufficient pumping force of the implanted heart [80].

Despite many recellularization and reendothelialization studies that have been conducted since 2008, none of the transplanted recellularized whole hearts showed functionality and long-term perfusion [71,79,111,112,113]. Due to the limited contraction in specific places of the left ventricle, reseeding cells could not develop a functional heart [106].

## 8. Cardiac Patch

Since gene and cell therapy did not provide a proper option for cardiac disease treatment and the transplantation of engineered whole hearts is beyond our reach with the existing facilities [3], a better approach would be to focus on repairing injured hearts and restoring cardiac function through the cardiac patch [2]. Eschenhagen et al., for the first time in 1997, introduced the concept of an in vitro CM-populated matrix using collagen hydrogel and embryonic chick CMs [114]. The term engineered heart tissues (EHTs) from neonatal rat heart-derived CMs was first proposed by Zimmermann et al. as the most relevant system for mammalian heart repair. The contractile force was measured, leading to a positive force–length and a negative force–frequency relationship [115].

The viable transplanted cells that interact with optimal electrophysiological and mechanical characteristics will make the ideal engineered cardiac tissue. The synthetic biodegradable materials that were studied are polymer poly(vinyl alcohol) (PVA), poly(lactic-co-glycolic) acid (PLGA), poly-(L-lactic) acid (PLLA), and polyurethanes (PU) [116,117]. There are many challenges with synthetic biomaterials, including inflammatory responses, mismatched material properties, insufficient bio-integration with recipients, and difficulty in monitoring the degradation rate of the biomaterials.

Currently, the most widely used biomaterial for the cardiac patch is decellularized ECM generated from cardiac or non-cardiac sources. Figure 1 shows the impact of decellularization and recellularization protocols on the cardiac extracellular matrix (ECM) [118]. Studies confirmed that agrin has a crucial role in promoting heart regeneration, which helps with the dystrophin-glycoprotein complex and subsequently promotes the in vitro division of CMs [119]. The most common cell sources for patch recellularization include umbilical cord stem cells, fetal CMs, induced pluripotent cells, and dermal FBs [116].

### 8.1. Small-Scale Natural Myocardial ECM Patch

#### 8.1.1. Development of Natural Myocardial ECM Patch in a Rodent Model

Chamberland et al. seeded mouse embryonic ventricular cells and mouse ESC-derived progenitors into the embryonic decellularized cECM, which resulted in the differentiation of ESCs into the cardiac, endothelial, and smooth muscle cells. Fabricated patches with embryonic ventricular cells started spontaneously beating after 20 h and those with ESC-derived progenitors started beating 24 days after seeding [120]. Lee et al. confirmed that a high proliferation rate results from the culture of neonatal rat CMs on thin decellularized cECM (10 µm thickness) sections [121]. Silva et al. cultured cardiac progenitor cells and neonatal CMs on fetal and adult decellularized cECM scaffolds, which showed better migration and colonization rates for fetal scaffolds [122]. The decellularized cECM from a donor SD rat supported the maturation of human iPSC-derived cardiac cells after reseeding. This construct showed normal electrical properties in response to the pharmaceutical agents in vitro. After grafting this patch on the acute rat MI model, the recellularized decellularized cECM reduced the infarct size, increased the wall thickness, and promoted vascularization. With respect to the high volume of patients who are seeking individual-specific human cardiac patches for the personalized regenerative therapy of myocardial infarctions, this kind of patch looks functional [123].

Hong et al. highlighted the increasing demand for post-myocardial infarction treatment over the coming years and offered skeletal ECM (sECM) as a substitute for engineered cardiac tissue (ECT) strategies since the microstructure and morphological properties of sECM were similar to the decellularized cECM. The sECM granted the adherence, survival, proliferation, and differentiation of murine embryonic stem cells into the functional cardiac microtissue with both stimulated electrical responses and normal adrenergic responses, which showed synchronized contractions within six days of repopulation [124]. The whole rat heart was decellularized by Hochman-Mendez et al., and 1.5 mm^2^ sections of the left atrial and left ventricular were cultured with human embryonic stem cell-derived CMs. It was observed that the mechanical properties, electrical activity, and cellularization were almost similar to native tissue [125]. Table 4 describes the rodent natural myocardial ECM used for the cardiac patch in previous publications.

#### 8.1.2. Development of Natural Myocardial ECM Patch in Porcine Model

Wang et al. seeded the mixture of undifferentiated and differentiated bone marrow mononuclear cells into decellularized myocardial slices with a 2000 µm thickness of the porcine heart. The results showed a proper reendothelialization with sufficient viability. The reseeded cells kept their cardiomyocyte-like phenotype, proven by immunohistological staining. By means of recellularization, a stiffer mechanical response of decellularized myocardial scaffolds was recovered [118].

Eitan et al. decellularized porcine myocardium tissues with a 3000 µm thickness and first reseeded with sheep cardiac fibroblast, resulting in scaffold shrinkage and ECM remodeling. Reseeded scaffolds with neonatal rat CMs started beating a few days after initial seeding, and functional cardiac markers such as a-actinin, troponin I, and connexin43 were expressed. The viability of seeded rat bone marrow-derived MSCs took up to 24 days in culture [127].

Sarig et al. found that the perfusion method with Triton X-100 and Trypsin is the most effective protocol for efficient cell removal while maintaining the structure of ECM, achieving long-term cell survival of the rat MSCs [128]. Some groups used decellularized cardiac patches without a recellularization process, such as the study by Wainwright et al., which showed that, in the regeneration of the right ventricular outflow tract (RVOT) in a rat model, a porcine acellular cardiac patch had better functional and structural outcomes when compared to a Dacron patch [128]. Another study demonstrated that Shah et al. decellularized two different thicknesses (300 and 600 μm) of porcine myocardial sections and implanted them on the rat myocardium’s infarcted area, which showed notable improvements in cardiac function and LV contractions four weeks after implantation. Porcine myocardial sections, when implanted in acute and chronic rat MI models, can be vascularized and promote constructive M2 macrophage phenotype. The CECM-supported recruiting progenitor (GATA4+, c-kit+) and myocyte (MYLC+, TRPI+), after implanting. The recruited progenitors expressed both early and late cardiomyocyte (CM) differentiation markers [129].

The acellular myocardial scaffolds with a 3000 µm thickness were repopulated with rat MSCs and kept culturing in the presence of 5-aza for 24 h. Two days after culturing with coordinated mechanical (20% strain) and electrical stimulations (5 V, 1 Hz), the scaffold was covered by viable cells. The immunofluorescence staining showed the differentiation of MSCs to a CM-like phenotype that can express sarcomeric α-actinin, myosin heavy chain, cardiac troponin T, connexin-43, and N-cadherin [130]. Perea-Gil et al. compared two decellularization protocols: detergent-based (SDS and Triton X-100) and Trypsin and acid with Triton X-100. There were no significant differences between the two developed myocardial scaffolds in the cell removal, preserving the ECM component and biomechanical properties. However, when the scaffolds were reseeded by adipose tissue-derived progenitor cells, the receded scaffold from protocol 1 contained a higher cell density that could express endothelial and cardiomyogenic markers [131].

Blazeski et al. used decellularized rat or pig ventricular sections 300 µm- thick to recellularize with neonatal rat ventricular cells, resulting in electrically paced functional tissue [132]. Ye et al. described the complete removal of cells from porcine myocardial slices with a 2000 µm thickness using the SDS-based method with an appropriate growth response in further experiments. In comparison with other dPMS decellularized with Trypsin and Triton X-100, the SDS-based method demonstrated a better preservation of ECM structure with better cell survival, growth, proliferation, and differentiation of the rat myocardial fibroblast and rat neonatal CMs after reseeding. Regarding beating magnitudes, the Trypsin-treated tissue had the largest, the SDS-treated tissue had a modest projection magnitude, and there was no beating in the Triton-treated group. The detergent-based method demonstrated a better preservation of the ECM structure with better cell survival, growth, proliferation, and differentiation of stem cells after reseeding [133]. Shah et al. compared the number of transplanted rat and pig ASCs in the infarcted area between delivering the cells through direct injection or repopulated thin dPMS. After one week, a higher number of transplanted cells were presented in the infarcted area when the patch was implanted into the infarcted rat heart than with direct injection, which led to an increased vascular formation within the patch. Compared with the MI group, the increased neovascularization was observed in the patch group as well [134].

Table 5 shows porcine and ovine natural myocardial ECMs used for the cardiac patch.

#### 8.1.3. Development of Natural Myocardial ECM Patch from Human Tissue

Oberwallner et al. decellularized 300 µm thick human and porcine myocardial slices with a protocol consisting of 2 h lysis, 6 h SDS, and 3 h FBS. Although the fabricated ECMs were intact for developing reproducible, high-quality scaffolds, the age and pathology of the donor tissue are highly important. The repopulated ECM with human umbilical cord blood-derived MSCs, murine iPSC-derived CMs, and murine neonatal CMs showed a higher viability and cell attachment for MSCs than the iPSC-derived CMs on the human dECM scaffold. Likewise, after seeding the neonatal CMs onto the matrix, their beating can move whole cardiac slices [142].

Meglio et al. examined five decellularization protocols. From these protocols, the best result was detected from the one that involved 1% SDS and 1% Triton for 24 h. The gene expression for SMCs and CMs increases after the culture of human cardiac primitive cells on the dECM scaffold with a 350 µm thickness [143]. Oberwallnera conducted another impressive research to prove the existence of cross-talk between ECM and stem cells. A different matrix, such as decellularized ECM and Matrigel or Geltrex, was tested for its impact on pluripotent stem cell differentiation. A 300 µm-thick decellularized cECM prepared from human myocardium can promote cell attachment, viability, proliferation, and cardiac lineage commitment of the seeded ESCs and iPSCs. The positive immunohistochemistry staining demonstrated this for cardiac troponin T and the heavy-chain cardiac myosin. Moreover, the mRNA expression for myosin-heavy polypeptide 6, cardiac troponin T2, and NK2 homeobox 5 was significantly increased. Matrigel or Geltrex could not induce cardiac-specific markers, unlike the dECM. Moreover, there is no evidence to show that MSCs are differentiated from CM [144].

Garreta et al. used hPSC-derived cardiomyocyte-like cells on an acellular ECM (400 µm thick) of the human cardiac scaffold, demonstrating positive immunofluorescence staining for alpha-sarcomeric actinin, Troponin T, MYH6, NKX2.5, and CX43 after ten days of cell seeding [145]. Table 6 provides a summary of natural human myocardial ECMs used for the cardiac patch.

#### 8.1.4. Drawbacks of Small-Scale Natural CECM Patch

Despite the recent advancements in cell delivery methods, only 0.1% to 10% of injected cells remain until a few hours after the transplantation [146]. Compared with the direct injection of cells into the myocardium, the use of small-scale engineered cardiac tissue results in a ten-fold higher engraftment rate [109,147,148]. The low engraftment rate still remains, and it is a critical issue that can limit the treatment’s effectiveness. To overcome the low engraftment rate, the fabrication of large and thick, clinically-sized cardiac patches may be a solution.

Within the first few hours after the transplantation, delivering oxygen to the engineered heart tissue is the main challenge, especially for tiny patches [149]. The average in vitro cardiac construct thickness was estimated to be about 200 µm [150]. The intercapillary distances of the native cardiac tissue were about 20 µm [151]. One of the main challenges is scaling up the in vitro cardiac tissue for clinical application.

The other hurdle of small-scale natural cECM patches is the mechanical conditioning for cell differentiation. The cardiac function is mainly monitored by passive myocardial stiffness. CM rhythmic contractions, which are induced by mechanical stress, pose a direct impact on force generation [1]. Small cardiac patches might not provide sufficient mechanical or structural support to cells compared with the larger patches.

### 8.2. Large-Scale Natural Myocardial ECM Patch

To date, there are many hurdles regarding the recellularization of large-sized whole heart samples due to the quantity and complexity of cells. Thus, downscaling the whole-heart samples into the thick decellularized matrices of ventricular slabs is likely to decrease the quantity and complexity of cells and enable more feasible long-term experimentation with bioreactors [135,137]. Figure 2 shows the interplay between decellularized cECM patch, growth factors, and further seeding of stem cells in small and large-scale cardiac patches.

Schulte et al. developed a novel ECM scaffold enriched with microvascular networks and biocompatible cell niches by decellularization of dissected left-anterior ventricular myocardium and its associated coronary arteries and veins. Rat dermal FBs were seeded onto lyophilized 5000 µm-diameter punch biopsies taken from myocardial flap scaffolds. Following the CMs seeded on dECM with diameters of 1000 µm, cells started expressing cardiac markers: sarcomeric α-actinin, myosin heavy chain, actin, and connexin43. These constructs are feasible to match the size of the infarct, integrate with the host myocardium, and be nourished by the host’s vasculature through a direct anastomotic connection. [137]. Sarig et al. co-cultured hMSCs and HUVECs on thick porcine myocardium dECM with 15,000 µm thickness by using a perfusion bioreactor chamber which led to functional vascularization/angiogenesis and synchronous beating of CMs [138]. Their team developed a 3D scaffold with an intact microstructural architecture with the flourishing perfusion-based decellularization method of coronary-based right heart flaps. Under dynamic culture conditions using a perfusion bioreactor, the decellularized cardiac flaps were repopulated with GFP+ rat neonatal cardiac cells. Confluent coverage of fibroblast, cardiomyocyte, endothelial, and smooth muscle cells exhibited on the recellularized myocardial flap were illustrated by positive immunohistochemistry staining for CD34, Desmin, α-SMA, Vimentin, connexin43. This result may pave the road for using an engineered viable patch by coronary artery bypass graft (CABG) to be implanted over a region of infarcted cardiac tissue without resecting the aneurysm scar tissue, which can reduce the risk of surgery. Using these pre-seeded scaffolds is promising for preventing ruptures in the left ventricle aneurysm following MI [141].

## 9. The Role of Cardiac Tissue Engineering in Clinic

Treatment of the left ventricular remodeling following MI is still seen as an unsolvable problem. Left ventricular reconstructive surgery is an option for LVA treatment, but high perioperative morbidity and mortality rates are always by its side [152]. Claude S. Beck introduced the surgical method for LVA repair in 1944 by using fascia lata aponeurosis to strengthen the wall of the LVA to reduce excessive stretching and avoid LV rupture [153]. However, this method was not feasible later. In 1955, Likoff and Bailey’s presented a new technique for LVA treatment; for LVA resection, they put a large vascular clamp on the beating heart tangentially across the base of an LVA [154]. To repair a postinfarction left ventricular aneurysm, several LV reconstruction methods were presented and divided into two techniques: the direct suture and the patch ventriculoplasty techniques. The most popular technique used nowadays was described by Levinsky et al. in 1979. Following resection of an anterior postinfarction aneurysm Dacron patch was sutured to the defect region [155]. For reconstructing the LV aneurysm some materials such as Dacron, bovine pericardium and Teflon are used as a patch.

Over and above, ventricular reconstructive surgery is a high-risk surgery, and cardiac TE has recently gained massive attention because of its application in the regeneration of damaged tissues, which leads to the development of novel therapeutic approaches for the treatment.

Clinical use of acellular bioscaffold was reported for the first time by Svystonyuk et al. The decellularized porcine small intestinal submucosal ECM (SIS-ECM) with preserved bioactive properties was cut into 1 cm diameter circles using a biopsy punch. For repopulating the acellular bioscaffold, the isolated hcFBs (~2500 cells per construct) were taken from the right atrial appendage or the left ventricular core of patients and seeded on the 1.0 mg/mL concentration of type I bovine dermal collagen poured onto the six 1-mm diameter holes that were made before in the bioscaffold. Later the seeded acellular bioscaffolds were implanted in the damaged areas within four weeks of ischemic injury through coronary artery bypass graftings in eight patients with MI. The perfusion in infarcted myocardium became better, the myocardial scar burden was reduced, the reverse remodeling was promoted, and the clinical outcomes were improved [156]. It seems Dacron is likely to be replaced by an acellular bioscaffold soon as some researchers showed the priority of biologic scaffolds for myocardial repair since they have structural and functional benefits over synthetic materials such as Dacron [128].

## 10. Summary and Future Directions

The high mortality and morbidity of cardiovascular diseases have remained a considerable health problem worldwide. Although multiple approaches have been developed and are not applicable to MI, heart transplant has remained the only definitive treatment method. As an immediate result of the loss of blood flow to the myocardium, inflammatory reactions start. Subsequent to inflammation, the functioning tissue of the heart will be replaced with non-functional fibrotic tissue. The collagen fibers’ structure will be affected, disrupting the ECM structure. These events cause dysfunction in the left and right ventricles [157].

The heart is a complex organ composed of specialized cells integrated into a scaffold that is rich with growth factors and sugars that provide biological signals for controlling cell behavior and a complex vasculature to oxygen and nutrient diffusion for cells [3]. From childhood to adulthood, the size of the human heart doubles. Additionally, as humans age, due to elasticity reduction in the heart muscles, the heart starts to thicken and stiffen, which affects organ shape and function; therefore, one of the characteristics that are necessary for the implanted patches is being able to adapt and remodel as the heart ages. Natural myocardial ECM patches can mimic the characteristics of a native healthy myocardium, and they can integrate into the host myocardium. The human heart comprises endothelial cells, CMs, FBs, smooth muscle cells, and specialized conducting cells, including pacemakers and Purkinje fibers. An adult human heart consists of approximately four billion CMs [108], which means we need billions of different types of cells for cardiac regeneration.

CMs are not applicable to cardiac tissue engineering; on the one hand, the myocardial biopsy is an invasive procedure, and the number of isolated cells will be insufficient. On the other hand, shortly after the isolation of adult human CMs from myocardial biopsies, the cells go through significant structural and functional remodeling, leading to cell dedifferentiation and loss of viability [158]. All these facts tell us that the use of CMs can be replaced by other cells, such as pluripotent stem cells, due to their high self-renewal capacity and ability to give rise to all cells of the human body. Pluripotent stem cells can either be derived from the inner cell mass of the preimplantation embryos, called embryonic stem cells [159], or by using in vitro technology forced expression of pluripotency genes through the delivery of the “transcription factors” Oct3/4, Sox2, Klf4, and c-Myc to somatic cells, which could reprogram them to be induced PSCs [160]. Induced PSCs can be expanded indefinitely, and they are able to differentiate into derivatives of all three germ layers [161]. Since ethics and safety are the major concerns associated with human embryonic stem cell transplantation, easy accessibility and using own patient-specific induced PSCs highlighted the role of the induced PSCs in the field of cardiac TE [162]. Differentiation protocols to achieve every subtype of CM (ventricular, atrial, and pacemaker) from PSCs with a high degree of purity were introduced [33], but still, these cells are much the same as fetal rather than adult CMs after the differentiation process. Since the immaturity of hPSC-CMs influences their application in different branches of medicine, having knowledge of cardiac development can help us to have a better view of relevant players that participate in the conversion of the fetal heart’s development into a functional organ. Several environmental elements, such as mechanical forces, electrical stimuli, biochemical factors’ gradients, ECM remodeling, and heterotypic interactions during cardiac development, are involved in CM maturation [34]. Some of those events begin in utero and the full development of adult CMs with the specialized cellular system will continue years after birth [163]. Therefore, even though the culture time of hPSC-CMs extended up to one year these cells still show signs of a fetal phenotype [164]. So, there is one chance left, which is trying to grow biomimetic cardiac tissue in the laboratory by moving from simple beating cells to engineering the physiologically relevant tissues.

Several bioengineering methods have been suggested to mimic native tissue microenvironment in vitro in order to promote hPSC-CM maturation, and decellularization is the most promising technique adapted to date. However, the idea of a whole-heart decellularization/recellularization failed due to other reasons. First, a large number of cells are required to occupy the entire volume of the whole decellularized heart. Second, all bioreactors are designed for single-organ; thus, bioreactors cannot offer complex interactions between the organs exposed in the human body [165]. Third, to avoid thrombosis formation, whole decellularized hearts have been successfully recellularized with endothelial and heart muscle cells produced contractility in vitro, but when these hearts were transplanted, due to the incomplete endothelialization potential, thrombosis occurred in the absence of anticoagulants, and when anticoagulants are used, the hemorrhage happened [80]. Fourth, none of the transplanted recellularized whole hearts are functional, nor do they have long-term perfusion since the pumping force of the transplanted heart is not sufficient to maintain life; only limited contraction was seen in specific places of the left ventricle [79].

Since the transplantation of engineered whole hearts, with our current knowledge and facilities, is beyond our reach, it would be a better approach to focus on repairing infarcted hearts and restoring cardiac function through the cardiac patches. The ideal engineered cardiac patch provides optimal structural, mechanical, and electrophysiological properties to keep the differentiated cells alive and promote the development of vasculature, supplying oxygen and nutrients in the patch region. As long as a small number of cells are delivered by tiny patches, the limited structural/mechanical means of supporting the cells will restrict the use of these kinds of patches [166]. Large and thick, clinically sized cardiac patches have become a feasible option for MI treatment, but one of the main challenges in scaling up the patches for clinical application is the delivery of oxygen and nutrients to the cells. Gao et al. confirmed that in the interiors of cardiac constructs with 2 mm thickness, which are kept under static conditions, oxygen concentration decreases to deficient levels, meaning only the tissue-engineered cardiac patch that is thinner than a few hundred micrometers can be implanted into the infarcted area. A large acellular myocardial flap with a coronary artery and its adjacent veins that match the size of the infarct and are nourished by direct anastomotic connection with the host’s own vascular network can be a good candidate. In that case, the proper number of cells delivered to the infarcted area and their viability and differentiation will be warranted. However, the cell viability, cellularity of the scaffold, clinical feasibility, and effectiveness of these methods are still under investigation.

For repopulating the patches, we need the collection of CMs as well as non-myocyte cardiac cells since CMs are never alone in the heart and there is always an interaction between them [167]. Adding different numbers of cardiac FBs (3%, 6%, and 12%) to ESC-derived CMs to fabricate the patches showed that the presence of cardiac FBs allowed the generation of synchronously contractile constructs while the patches consisting of just ESC-CMs failed to form a functional syncytium [168]. Another research work confirmed that adding non-myocytes to CMs can increase the survival and maturation levels of these cells [169]. After implanting repopulated patches, the integration of the myocyte in the graft with the rest of the myocytes in the heart is vital because the myocytes in the graft can act as pacemaker cells, causing an arrhythmia. Moreover, the difference in the speed of the cardiac electrophysiological signals of the graft and host myocardium can cause reentry arrhythmia [170]. PSCs show potential for cardiac regeneration as they are able to expand indefinitely in vitro and differentiate into multiple cell types, including CMs. Still, since the use of stem cells in the undifferentiated state is associated with the risk of forming teratomas in vivo, clinicians are hesitant to work with them [171]. More than all those complexities regarding working with the cells, the patches must be prepared freshly to maintain cell viability, so their clinical applicability will be limited due to the impossiblity of keeping them for long terms. Continuous research in the treatment of MI came up with new ideas and technologies to improve heart function.

Applications of exosomes have been accompanied by satisfying regenerative results. Exosomes derived from various stem cells have provided regenerative abilities, especially angiogenesis for damaged cardiac tissues [172]. After murine ESC exosomes were delivered to infarcted mouse hearts, angiogenesis and CM survival were promoted, fibrosis was decreased, and cardiac function was improved [173]. Moreover, miRNAs are supposed to play essential roles in repairing cardiac tissue damage. It has been demonstrated that some regenerative abilities of exosomes are being carried out through miRNAs, including miR-21–5p, from MSCs, which can raise a cardiac contractile force and calcium handling through regulating PI3K signaling [174].

As we mentioned before, clinical applications of stem cells for heart repair have several limitations; therefore, using their secretive instead of living iPSCs seems a safer option. The paracrine release of tumor necrosis factor-α (TNF-α), interleukin (IL)-8, granulocyte colony-stimulating factor, and VEGF can enhance cell engraftment and promote angiogenesis; they can also improve cell proliferation and inhibit apoptosis, resulting in the repair of the infarcted myocardium [175].

Fabrication of an off-the-shelf therapeutic cardiac patch by encapsulating the synthetic cardiac stromal cells into the decellularized ECM can repair the rat and porcine infarcted myocardium, and their long-term storage is feasible [176].

Cardiac patches fabricated by embedding PLGA-encapsulated synthetic cardiac stromal cells or exosome or paracrine into a decellularized myocardial scaffold would be a useful technique. Decellularized myocardial scaffold will provide a suitable physiological environment for biomolecular delivery. Paracrine factor release is an alternative to direct interaction with cells with host CMs that play a key role in cardiac repair [177]. The acellular patches integrated with paracrine factors showed more direct therapeutic impacts than the cellular ones [178]. Even if cardiac patches are fabricated with only decellularized ECM, they are capable of treating infarcted myocardium through mechanical–structural supports. These kinds of patches, without any external therapeutic agents, can protect the left ventricular from remodeling and stretching, and they have the potential to recruit progenitors expressed in both early and late CM differentiation markers [179].

All these patches are cell-free and clinically feasible. Moreover, they can replace long-term storage and myocardial cell-based cardiac patches.

## Figures and Tables

**Figure 1 bioengineering-10-00106-f001:**
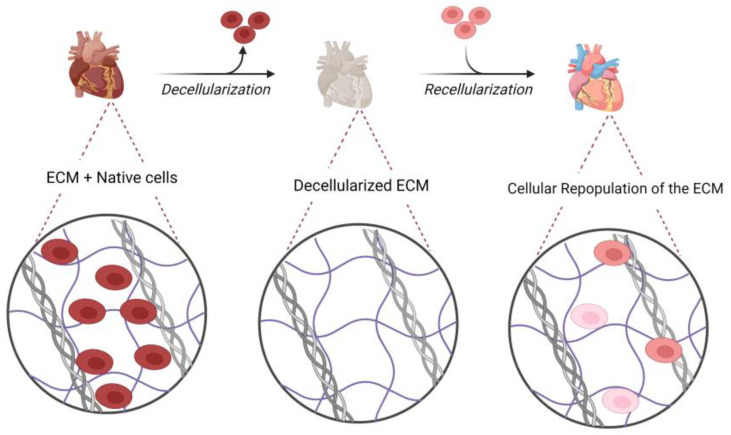
The impact of decellularization and recellularization protocols on cardiac extracellular matrix.

**Figure 2 bioengineering-10-00106-f002:**
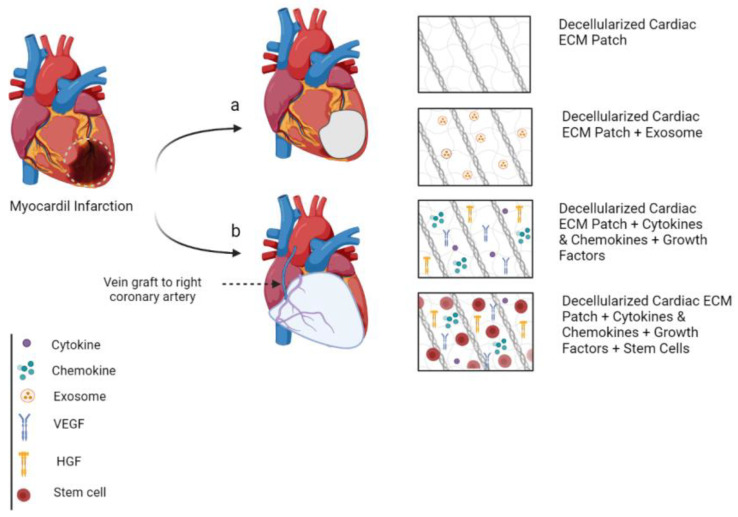
The interplay between decellularized cECM patch, growth factors, and further seeding of stem cells; (**a**) small patch, which cells will die because they do not have access to vascular network when they are implanted, (**b**) large-scale cardiac patch.

**Table 1 bioengineering-10-00106-t001:** Rodent whole-heart decellularization.

Animal Model	Decellularization Agent	Cell Source	Major Result
Rat [79]	1% SDS; 1% Triton-X100	Neonatal rat CMs; fibrocytes; endothelial cells; smooth muscle cells; rat aortic endothelial cells	Sacromeric-α-actin, cardiac α-myosin heavy chain, and Connexin-43 were expressed after the recellularization of the whole heart with cardiac cells. However, the implanted tissue-engineered heart produced only 2% of adult rat heart function, insufficient pumping for life.
Mouse [81]	1% SDS; 1% Triton-X100	Human embryonic stem cells (hESCs); human mesendodermal cells (hMECs) derived from hESCs	Differentiation of the hESCs and hMECs, which were differentiated from hESCs in the decellularized scaffold after recellularization, was approved by upregulated expression of cardiac markers such as cTnT and Nkx2.5 upon differentiation. Part of the stem cells was differentiated to CD31 positive cells. There is no beating in the recellularized scaffold.
Rat [82]	SDS; sodium deoxycholate (SDC); glycerol+ saponin	Mouse myoblast cell line (C2C12)	Introducing a new protocol and comparing it with three published protocols for rat whole-heart decellularization in terms of remaining residual DNA content, preservation of ECM, and viability of C2C12 myoblasts after reseeding into the ECM. Results showed that none of the scaffolds were entirely free of cellular components and could not keep the ECM structure.
Rat [83]	1% SDS	-	The mechanical characteristics of the right ventricles of the hearts were examined after the decellularization process, and the presented decellularized cECM had notably higher stiffness compared to the native myocardium tissue.
Rat [84]	1% SDS; 1% Triton-X100	Canine blood outgrowth endothelial cells	Rat hearts were decellularized and cryopreserved at −80 °C with 10% DMSO in PBS, and the recellularization process with canine ECs completed after 1 year. On day 9 of the culture, seeded cells in the lumen showed viability and well-attached morphology. Long intervals between decellularization and reseeding might be a predictor of viability for commercial scales.
Rat [71]	1% SDS; 1% Triton-X100	RAECs	This is the first study to re-endothelialize whole decellularized hearts through both arterial and venous beds and cavities. Scaffold vessel re-endothelialization with RAECs was increased in the combination of brachiocephalic artery (BA) + inferior vena cava (IVC) delivery strategy in comparison with single-route strategies.
Mouse [85]	0.02% Trypsin; 0.05% ethylenediaminetetraacetic acid (EDTA); 0.05% NaN3; 1% SDS; 3% Triton X-100; 4% deoxycholic acid (DCA)	Human iPSC-derived cardiac progenitor cells	The hearts beat on day 20 of the recellularization of a whole mouse heart with cardiac multipotent stem cells derived from human iPSCs (Y1-iPSCs). Differentiation into cardiac myocytes, smooth muscle cells, and ECs on the ECM was confirmed, and it was presented that ECM provokes the proliferation of cardiac myocytes.
Rat [94]	sodium lauryl ester sulfate (SLES); 1% TritonX-100; 500 U/mL DNase	-	A novel detergent, sodium lauryl ether sulfate (SLES), was used to compare with SDS for decellularization. The resulting ECMs from SLES-treatment were better preserved in comparison with ECMs from SDS-treatment. Mesenteric transplantation of both ECM revealed SLES did not elicit an immune response, unlike SDS.
Rat [86]	80% glycerol; 0.9% NaCl; 0.05% NaN3, 25 mM; EDTA, 4.2% sodium deoxycholate;1% SDS; 3% Triton X-100	Neonatal rat cardiac cells	Cardiac markers (α-actinin, cTnI, connexin 43), endothelial marker (vWF), and proliferation marker (ki67) were positive after reseeding whole decellularized rat hearts with neonatal rat cardiac cells. Within 2 days after repopulation, the heart began beating. This model only accounts for about 36% of the mass of an adult rat heart.
Neonatal mouse [87]	1% SDS; 1% Triton-X100	Neonatal murine cardiomyocyte	Reseeding of the decellularized neonatal mouse heart with neonatal murine cardiomyocyte showed positive cells for both NKX2.5 and α-actinin, expressed by cardiac progenitor cells and differentiated CMs, respectively. Even after these cells undergo prolonged culture continue to express cardiomyocyte markers. So, the neonatal matrices have enormous potential to be a novel construct for repopulation.
Rat [90]	1% SDS	-	Comparing two groups of rat hearts, one decellularized with pulsatile perfusion, the other with non-pulsatile perfusion showed significantly lower residual DNA content in the scaffolds developed by the pulsatile perfusion, which is attributed to the higher hemodynamic energy of pulsatile perfusion. There are no significant differences between the groups in collagen and GAG contents.
Rat [88]	1% SDS; endonuclease in Hank’s balanced salt solution (25 U/mL); ultra-pure water; collagenase IV solution	HEK293; primary human cardiac FBs; human iPS-derived cardiac progenitor cells (iPS-CPCs)	Four-Flow cannulation is adjusted for whole heart decellularization and recellularization through cannulating the ascending aorta (AA), superior vena cava (SV), pulmonary artery (PA), and pulmonary vein (PV). After the recellularization process, HEK293 cells seen around the vascular structures and seeded primary hcFBs were able to migrate and attach to the scaffolds. Growth of cell patches can be seen macroscopically when the h-iPS-CPCs were perfused to the scaffold.
Rat [91]	Protease inhibition; antioxidation; 0.5% SDS, 1% Triton-X100	-	By applying protease inhibition, antioxidation, and excitation−contraction uncoupling in simultaneous perfusion/submersion modality, as a novel decellularization method, detergent concentration and detergent exposure time were reduced.
Mouse [92]	1% SDS; 1% Triton-X100	Induced cardiac progenitor cells (iCPCs)	After 3 weeks of repopulation of the whole decellularized heart with iCPCs, cells began migrating, colonizing, and finally differentiating in a scaffold, as demonstrated by the detection of cardiac actin-positive CMs, SMA-positive smooth muscle cells, and CD31^+^. Electrically functional cardiomyocyte clusters have appeared in the scaffold under field stimulation.
Rat [88]	1% SDS	Human mesenchymal stem cells (hMSCs)	The detergent exposure time was decreased by using an electric field in the decellularization protocol. To repopulate the decellularized ECM, hMSCs differentiated iCMs with 5-azacytidine (5-aza), and these cells were similar to CMs and arranged in fibers.
Rabbit [93]	Hypertonic (500 mM) saline; hypotonic (20 mM) saline; 1% SDS	Human iPSC-derived ECs; hiPSCs-derived cardiac cells (CCs)	The whole decellularized rabbit heart was reendothelialized by ECs and recellularized by CCs. The cells (ECs and CCs) recovered 92.6% ± 18.4% of LV wall thickness after 60 days. Cardiomyocyte maturation was confirmed by decreasing the percentage of cells positive for progenitor markers. Recellularized hearts exhibited visible beating. Results revealed maintaining vessel patency after transplanting this heart to the femoral artery bed of a pig and starting perfusion.

**Table 2 bioengineering-10-00106-t002:** Porcine/ovine whole-heart decellularization.

Animal Model	Decellularization Agent	Cell Source	Major Result
Porcine [95]	0.02% Trypsin/0.05% EDTA; 0.05% NaN3 solution; 3% Triton X-100/0.05% EDTA; 4% DCA solution	White leghorn chicken embryonic CMs	Using pulsatile retrograde aortic perfusion for 10 h resulted in whole porcine decellularized heart with intact ECM component. Lyophilized cECM sheets were repopulated with 500,000 cells/cm^2^ white leghorn chicken embryonic CMs. The development of organized chicken cardiomyocyte sarcomere structure was supported by cECM.
Porcine [96]	4% SDS	-	The Langendorff perfusion decellularization model was improved by adding a heat exchanger to keep the temperature of perfusion fluids at 37 °C. A pressure transducer can control the roller pump through a computer system. The decellularized scaffolds produced by this system were cell-free, while their ECM components were preserved.
Porcine [103]	0.02% Trypsin/0.05% ethylenediaminetetraacetic acid; 0.05% NaN3 solution; 3% Triton X-100/0.05% ethylenediaminetetraacetic acid; 4% DCA solution	-	By increasing the perfusion speeds greater than 2 L/min, exposure time of the tissue to detergent was minimized. The loss of cardiac muscle bundles in decellularized scaffolds was confirmed, while ECM substances and micro-architecture remained.
Porcine [104]	Protocol I: 0.02% Trypsin/0.05% EDTA/0.05% NaN3 for 3 days and 3% Triton X-100/0.05% EDTA/0.05% NaN3 for 4 days; Protocol II: 0.02% Trypsin/0.05% EDTA/0.05% NaN3 for 1 day and 3% Triton X-100/0.05% EDTA/0.05% NaN3 for 6 days; Protocol III: 0.02% Trypsin/0.05% EDTA/0.05% NaN3 for 7 days; Protocol IV: 3% Triton X-100/0.05% EDTA/0.05% NaN3 for 7 days	-	Comparing three decellularization protocols showed that using Trypsin only and Triton only could not fully remove nuclear materials. In Triton only group, collagen network was saved more as same as elastin, which could contribute to the increase in the compressive modulus in this group. These data revealed that, to characterize the structural and mechanical properties of decellularized scaffolds, noninvasive and nondestructive methods, such as multiphoton microscopy (MPM) combined with image correlation spectroscopy (ICS), could be used.
Porcine [97]	4% SDS	Human umbilical cord- derived endothelial cells (HUVEC); neonatal rat cardiomyocyte	After reseeding the decellularized porcine hearts with neonatal rat cardiac cells and HUVEC, viable cardiac cells were detected, and coronary vasculature was reendothelialized. The electrical activity of the heart remained for up to 3 weeks in perfusion bioreactor system.
Porcine [98]	1% SDS; 1% Triton X-100	Porcine mesenchymal stem cells (pMSCs)	Two groups of whole porcine hearts: decellularized and recellularized with pMSCs, were transplanted into pigs under systemic anticoagulation treatment with heparin. On day 3, hearts from both groups were harvested. Porcine MSCs were not found in vessel lumen, and a coronary thrombosis was detected in both groups.
Porcine [99]	Hypertonic (500 mM NaCl); hypotonic (20 mM NaCl; 1% SDS	-	Porcine hearts were decellularized with two unique retrograde methods, inverting the heart and venting the apex. The inverted method offers a patent coronary vascular architecture with a higher coronary perfusion efficiency that can help ECM to rid of more native cells. Moreover, improved ECM preservation and correspondingly better retention of the heart shape following decellularization are also obtained from inverted method.
Ovine [101]	0.75% SDS; 0.25% DCA	Neonatal rat primary cardiac FBs	After whole-heart decellularization, total protein content in EAT, as regulator of cardiac anatomy and function, was strongly reduced, and large amounts of lipids were detected in EAT, indicated by lipid staining. However, there is no donor material in other regions of heart. Incubation cardiac FBs with decellularized EAT showed a significantly diminished viability versus incubating with native EAT or unconditioned culture medium. Overall, incomplete removal of donor material can cause cytotoxic effects.
Porcine [100]	0.5%. SDS; 1% Triton X-100	-	Detergent exposure time decreased by using automated pressure-controlled bioreactor. The decellularization process is standardized by helping automated systems and effective liquid perfusing through coronary arteries achieved by pressure-controlled system. Cell components of the whole porcine heart were completely removed (98%) with only 6 h of detergent exposure in this modified bioreactor.
Ovine [102]	1% SDS; 1% Triton-X100	-	After decellularization process of whole ovine heart, different parts of the heart, such as auricle, aortic valve, left and right ventricular myocardia and chordae tendineae, were examined separately due to having a variable composition of ECM. Collagen and GAGs contents in chordae tendineae were decreased. The microvascular angiography indicated the natural 3D architecture of coronary tree was preserved. Harvested, subcutaneously implanted dECM into the omentum of Sprague–Dawley (host) rat, after 2 months, showed good vascularization and repopulation of the graft, indicated by the existence of CD31+, CD34+, and smooth muscle cells.

**Table 3 bioengineering-10-00106-t003:** Human whole-heart decellularization.

Whole Heart	Decellularization Agent	Cell Source	Major Result
Human [105]	1%SDS	hCPC; human bone-marrow mesenchymal cells (hMSCs); HUVECs; H9c1 rat CMs; HL-1 CMs	After 21 days of repopulating of whole human heart with various cells, CMs genes were expressed by hCPCs and hMSCs, but they did not show cardiomyocyte morphology. Endocardium and vasculature were covered by HUVECs. Well-organized H9c2 and HL1 cardiomyocytes into nascent muscle bundles exhibited mature calcium dynamics and electrical coupling
Human [106]	1% SDS; 1% Triton-X100	CMs human BJ fibroblast RNA-induced pluripotent stem cells (BJ RiPS)-derived CMs; human-induced pluripotent stem cell-derived CMs	After recellularization of whole decellularized heart with human BJ RiPS-derived CMs, most of the seeded cardiomyocytes expressed cardiac markers, including sarcomeric α-actinin, cardiac troponin T, and MHC. Under biomimetic culture, the repopulated scaffolds showed electrical conductivity, developing left ventricular pressure, and metabolic function. Cadaveric and decellularized human myocardium were subcutaneously implanted into the Sprague–Dawley rats and harvested after 2 weeks. CD68+ mononuclear cells in 2 groups were detected. M2 macrophages appear in greater quantities in decellularized human myocardium in comparison with cadaveric humans in proinflammatory response
Human [107]	Hypertonic (500 mM NaCl); hypotonic (20 mM NaCl; 1% SDS	-	In keeping the aortic valve closed to improve myocardial decellularization perfusion system, pressurized pouch can help by generating pressure gradients across the aortic valve

**Table 4 bioengineering-10-00106-t004:** Rodent natural myocardial ECM used for cardiac patch.

	Decellularization Agent	Cell Source	Major Result
Mouse [120]	0.25% SDS; 0.5 mg/mL DNase	Mouse embryonic ventricular cells; mouse ESC-derived progenitors	To optimize the differentiation and maturation of embryonic stem cells, mouse embryonic ventricular cells, and mouse ESC-derived progenitors were seeded on embryonic decellularized cECM, and ESCs were differentiated, proved by expression of endothelial, cardiac, and smooth muscle markers causing spontaneous beating after 20 h and 24 days of culture, respectively.
Rat [121]	0.25% Triton X-100/10 mmol/l NH4OH	Neonatal rat CMs	Seeding neonatal rat cardiac cells on decellularized heart scaffold sheets with 10 µm thickness produced higher proliferation rates, cardiac genes, and protein expression compared with those cultured without the ECM sheets.
Rat [126]	1% SDS; 1% Triton X-100	iPS	Seeding iPS cells on the decellularized cardiac scaffold confirmed proper cell scaffold attachment, cell survival, and cell differentiation and growth; furthermore, this phenomenon is characterized by decreased expression of pluripotency markers (Oct-4 and SSEA-4) after 7 days of culture.
Rat and mouse [122]	10 mM Tris HCl/0.1% EDTA; 0.2% SDS/10 mM Tris HCl; DNAse (50 U/mL)	Immortalized adult Lin-Sca1þ cardiac progenitor cells (iCPCSca-1); neonatal rat CMs	After repopulating both fetal and adult ECM matrices with cardiac progenitors and neonatal CMs, these cells migrated into the scaffolds with excellent viability. Fetal scaffolds showed better repopulation efficiency, migration, and colonization rates of seeded cells rather than adult ECM matrices.
Rat [123]	1% SDS; 1% Triton X-100/ 0.5% EDTA	Human-induced pluripotent stem cell-derived CMs; human-induced pluripotent stem cell-derived CD90^+^ cells	Decellularized cECM supported the maturation of human iPSC-derived cardiac cells in vitro. This construct showed normal electrical properties in response to the pharmaceutical agents. After grafting this patch on the acute rat MI model, the recellularized decellularized cECM reduced the infarct size, increased the wall thickness, and promoted vascularization.
Mouse [124]	0.05% Trypsin/0.02% EDTA, 1.1% NaCl, and 0.7% NaCl; 0.1% SDS; 1% Triton X-100	Murine embryonic stem cells (mESC); murine embryonic stem cell-derived CMs	Shortage of organ donation and immunogenicity offered skeletal ECM (sECM) as a substitute for engineered cardiac tissue (ECT) strategies since the microstructure and morphological properties of sECM were similar to decellularized cECM. SECM granted the adherence, survival, proliferation, and differentiation of murine embryonic stem cells into functional cardiac microtissue with both stimulated electrical responses and normal adrenergic responses, which showed synchronized contraction within 6 days of repopulation.
Rat [125]	1% SDS; 1% Triton X-100	Human embryonic stem cell-derived CMs	Isolated decellularized left atrial (LA) and decellularized left ventricular (LV) cECM were repopulated by human embryonic stem cell-derived CMs. The myoglobin levels in the recellularized LA and LV were almost 57% and 55%, respectively, of the level in cadaveric heart tissue; this confirmed a four- to five-fold increase in myoglobin levels in both the rLA and rLV than the level in hESC-CMs cultured alone. Both tissue groups presented synchronous depolarization in non-adjacent regions, and connexin 43 was expressed by CMs. The elastic properties of the dECM scaffolds were restored, similar to that in cadaveric tissues.

**Table 5 bioengineering-10-00106-t005:** Porcine and ovine natural myocardial ECM used for cardiac patch.

Animal Model	Decellularization Agent	Cell Source	Major Result
Porcine [118]	0.1% SDS/0.01% Trypsin; 1 mM phenylmethylsulfonylfluoride; 20 mg/mL RNase A/0; 2 mg/mL DNase	Porcine bone marrow mononuclear cells	After decellularization of myocardial slices with 2000µm thickness, the scaffolds were cultured with a combination of undifferentiated and differentiated bone marrow mononuclear cells toward cardiac phenotype. Attachment, viability, infiltration, and proliferation of seeded cells were guaranteed by dECM. Cardiomyocyte-like phenotype was maintained, and possible endothelialization within the scaffold was observed. By means of recellularization, stiffer mechanical response was recovered.
Porcine [127]	1.1% NaCl/0.02% EDTA and 0.7% NaCl/0.02% EDTA; 0.05% Trypsin/ 0.02% EDTA; 1% Triton X-100 and 0.1% ammonium hydroxide	Sheep cardiac fibroblast; neonatal rat cardiac myocytes; rat bone marrow-derived MSCs	Porcine myocardium tissues with 3000 µm thickness were decellularized and seeded with cardiac fibroblast, resulting in the scaffold shrinkage and ECM remodeling. Reseeded scaffolds with CMs started beating few days after initial seeding, and functional cardiac markers were expressed. The seeded MSCs remained viable for 24 days in culture.
Porcine [135]	1.1% NaCl and 0.7% NaCl; 0.05% Trypsin/0.02% EDTA; 1% Triton X-100/ 1% ammonium hydroxide	Rat MSCs; human umbilical vein endothelial cells	Left ventricular tissue slabs containing the LAD coronary artery and its adjacent veins were dissected. Trypsin/Triton-based perfusion procedure resulted in a nonimmunogenic and cell-supportive dECM, which was found to be more effective than stirring, sonication, or sodium dodecyl sulfate/Triton-based procedures to achieve thick dECM. The dECM scaffold with 14,600 ± 1900 µm thickness supported the attachment and long-term cell survival of rat MSCs. Monolayer of HUVECs was formed in the inner lumen of the intact vasculature of dECM.
Porcine [136]	0.1% SDS/0.01% Trypsin;1 mM phenylmethylsulfonylfluoride; 20 mg/mL RNase A/0; 2 mg/mL DNase	-	By using supporting system and a rotating bioreactor, treated porcine myocardium slices with 0.1% SDS and 0.01% Trypsin solution by a frame-pin resulted in dECM scaffold with 2270 ± 380µm thickness, which is free from cells and α-Gal porcine antigens. Decellularized ECM showed stiffer tensile properties than native porcine myocardium tissue.
Porcine [128]	0.02% Trypsin/0.05% EDTA/0.05% NaN3; 3% Triton X-100/0.05% EDTA/0.05% NaN3; 4% deoxycholic acid	-	Comparing porcine decellularized ECM cardiac patch with Dacron patch for reconstruction of a full-thickness RVOT in a rat model showed that the Dacron patch was encapsulated by dense fibrous tissue and a few cell infiltrations while the decellularized cECM patch remodeled into dense, cellular connective tissue that spread small islands of CMs, leading to improvement in heart function.
Porcine [130]	0.1% SDS/0.01% Trypsin; 1 mM phenylmethylsulfonylfluoride and 20 mg/mL RNase A/0.2 mg/mL DNase	Rat MSCs	Porcine myocardium tissues with 3000 µm thickness were decellularized and reseeded with rat MSCs and exposed to 5-aza treatment. After 2-day culture with coordinated mechanical (20% strain) and electrical stimulation (5 V, 1 Hz), MSCs differentiated to CM-like phenotype can express sarcomeric α-actinin, myosin heavy chain, cardiac troponin T, connexin-43, and N-cadherin.
Porcine [137]	30 mM EDTA; 1% SDS; 0.1 M NaOH	Rat dermal FBs; neonatal rat CMs	After decellularization of dissected flap with a 10,000 µm thickness, 5000 µm-diameter punch biopsies were taken and lyophilized. Rat dermal FBs were seeded onto the dry scaffold samples; they showed high cell viability. Following CMs seeded on dECM with 1000 µm-diameter, cells started expressing cardiac markers: sarcomeric α-actinin, myosin heavy chain, actin, and connexin43.
Porcine [131]	Protocol 1: 1% SDS; 1% Triton X-100; 0.1 mg/ ml DNase I Protocol2: 1.1% NaCl and 0.7% NaCl; 0.05% Trypsin/0.02% EDTA; 1% Triton X-100/ 1% ammonium hydroxide	Adipose tissue-derived progenitor cells (ATDPCs)	Porcine myocardium tissues with 3000 µm thickness were decellularized with two decellularization protocols: 1. detergent-based and 2. Trypsin and acid with Triton X-100. After one week of reseeding dECM with ATDPCs, receded scaffold from protocol 1 contained higher cell density, and differentiated cells expressed the cardiac markers GATA4, connexin43, and cardiac troponin.
Rat and porcine [132]	SDS; Triton X-100	Neonatal rat ventricular cells (NRVCs)	Rat or pig ventricular tissue was sectioned into 300 µm-thick slices, decellularized, spread on coverslips, and reseeded with NRVCs. The ECM can promote cell elongation and alignment, resulting in fabrication of an anisotropic, functional tissue that could be electrically paced.
Porcine [138]	1.1% NaCl and 0.7% NaCl; 0.05% Trypsin/0.02% EDTA; 1% Triton X-100/ 1% ammonium hydroxide	Bone marrow-derived MSCs; human umbilical vein endothelial cells; human ESC-derived CMs	Comparing the repopulation of thick decellularized porcine myocardium with 15,000 µm thickness by mixture of hMSCs and HUVECs using a perfusion bioreactor or static culture conditions showed higher cell infiltration through bioreactor, which can fabricate functional vascularized construct. Three days after initial human ESC-derived CMs seeding, synchronous beating was observed.
Porcine [139]	1.1% NaCl/0.02% EDTA and 0.7% NaCl/0.02% EDTA; 0.05% Trypsin/ 0.02% EDTA; 1% Triton X-100 and 0.1% ammonium hydroxide	-	Porcine myocardial sections with 1500 µm thickness were decellularized and implanted in acute and chronic rat MI models. In both MI models, cECM were vascularized and induced a constructive remodeling process, as explained by increased M2/M1 macrophage phenotypic ratio. CECM promoted recruiting progenitor (GATA4^+^, c-kit^+^) and myocyte (MYLC^+^, TRPI^+^) after implanting. Recruited progenitors expressed both early and late CMs differentiation markers.
Porcine [132]	1% Triton X-100, 1% SDS, and 0.5% Trypsin	Rat myocardial fibroblast (rMFs); rat neonatal CMs	Porcine myocardial slices with 2000 µm thickness were decellularized with different protocols. Compared to decellularization with Trypsin and Triton X-100, the SDS-based treatment has resulted in better ECM with the preserved component and microstructure. Reseeding of three different decellularized scaffolds with rMFs and rCMs showed the following: in SDS group, rMFs appeared to be more uniformly aligned. The long axis of the CMs in the SDS group tended to align in parallel, which contrasts the random orientation in Trypsin group. Regarding beating magnitudes, the Trypsin-treated tissue had the largest, the SDS-treated tissue had a modest projection magnitude, and there was no beating in the Triton-treated group.
Porcine [133]	10 mM Tris/0.1% EDTA; 0.5% SDS; DMEM containing 10% fetal bovine serum; 0.1% peracetic acid/4% ethanol	Neonatal rat ventricular myocytes (NRVMs); human ESC-derived CMs; human-induced pluripotent stem cell-derived CMs (hESC-CMs)	After decellularization of thin myocardial sheets (150 µm thickness) prepared by laser cutter, NRVMs seeded on the decellularized porcine myocardium slice (dPMS) produced synchronously beating scaffolds and showed a striated pattern of organized sarcomeres. Another group that was seeded with hESC-CMs resulted in beating scaffolds with measurable intracellular calcium transients and maximum twitch stress of 1.7 N/mm^2^. Likewise, seeded dPMS with hiPSC-CM reached maximum peak stress of 6.5 mN/mm^2^ and twitch kinetics similar to the previously reported values regarding adult human right ventricular trabecula.
Porcine [140]	1% SDS; 0.01% Triton X-100	Human mesenchymal stem cells (hMSCs); rat adipose-derived stem cells(rASCs)	Cardiac cryosections were taken in different sizes (300, 600, and 900 μm thickness) and decellularized. When hMSCs and rASCs were cultured on top of the dPMS from one side (lateral cell seeding) and both sides (bilateral cell seeding), these dPMS supported cell attachment with high viability and induced endothelial differentiation and maturation of hMSCs and rASCs after 1 and 5 days. However, seeded bilateral 600 μm dPMS group could significantly enhance seeding efficiency, infiltration, and growth of cells.
Porcine [134]	1% SDS; 0.01% Triton X-100	Rat Adipose-derived stem cells (rASCs); pig adipose-derived stem cells (pASCs)	Thin layers (300 µm thickness) of decellularized porcine myocardium repopulated with rASCs and pASCs. Both rat and pig ASCs showed high viability and similar patterns of proliferation and infiltration within the dPMS. ASCs were delivered to the infarcted myocardium (rat model) using dPMS. After 1 week, a higher number of transplanted cells were presented in the infarcted area than in direct injection, which led to increased vascular formation within the patch.
Porcine [129]	1% SDS	-	Two different thicknesses (300 and 600 μm) of dPMS were patched into the infarcted rat heart. After implantation, implanted dPMS was strongly attached to host myocardium and prevented thinning of the left ventricular (LV). Neovascularization into scaffold began soon after implantation, and a large number of host cells were recruited into the implanted dPMS. In treated rats with dPMSs, higher density of M2 macrophages was observed as compared to the MI group without treatment. Four weeks after surgery, contraction of the LV wall and cardiac functional parameters made significant progress.
Ovine [141]	1% SDS	Green fluorescent protein (GFP^+^) rat neonatal cardiac cells	Right heart with attached pedicle and coroners was dissected and decellularized through perfusion-based method. Then, decellularized cardiac flaps were repopulated with GFP^+^ rat neonatal cardiac cells under dynamic culture conditions using a perfusion bioreactor. Confluent coverage of fibroblast, cardiomyocyte, endothelial, and SMCs exhibited on the recellularized myocardial flap were indicated by positive immunohistochemistry staining for CD34, Desmin, α-SMA, Vimentin, connexin43.

**Table 6 bioengineering-10-00106-t006:** Natural human myocardial ECM used for cardiac patch.

Animal Model	Decellularization Agent	Cell Source	Major Result
Human and porcine [142]	10 mM Tris/0.1% EDTA; 0.5% SDS	Human umbilical cord blood-derived MSCs; murine iPSC-derived CMs; murine neonatal CMs	Human and porcine myocardial sections with 300 µm thickness were decellularized and cell-seeded. Murine iPSC-derived CMs showed less cell attachment, proliferation, and infiltration on the human dECM compared to MSCs. In standard culture, murine neonatal CMs contracted synchronously as it continued after seeding onto the matrix, and their beating could make strong contractions to move the whole cardiac slices
Human [145]	10 mM Tris/0.1% EDTA; 0.5% SDS	Murine ESCs; murine-induced pluripotent stem cells; murine mesenchymal stromal cells	A 300 µm thick decellularized cECM prepared from human myocardium was compared with another matrix such as Matrigel or Geltrex to show their positive effect on differentiation of stem cells. Promoting cell attachment, viability, proliferation, and cardiac lineage commitment of seeded ESCs and iPSCs seeded on decellularized cECM was proved by positive immunohistochemistry staining for cardiac troponin T, heavy-chain cardiac myosin plus, mRNA expression for myosin heavy polypeptide 6, cardiac troponin T2, and NK2 homeobox 5 being significantly increased. Matrigel and Geltrex were not able to induce cardiac-specific markers. There is no evidence to show that MSCs differentiated from CM
Human [143]	1% SDS	Human ESC-derived CM-like cells; human-induced pluripotent stem cell-derived CM-like cells (CLC)	Human decellularized cECM with 400 µm thickness, when cultured with hPSC-derived CLCs, could lead to differentiation and maturation of hPSC-derived CLCs toward CMs, as illustrated by positive immunofluorescence staining for alpha-sarcomeric actinin, Troponin T, MYH6, NKX2.5, and CX43. Moreover, after 10 days of culture (as more evidence for differentiation of hPSC-derived CLCs), levels of expression of different ion channels determinant for calcium homeostasis and heart contractile function enhanced Significantly.
Human [144]	Protocol I: lysis buffer (10 mM Tris, 0.1% wt/vol EDTA, pH 7.4); 0.5% SDS Protocol II: 10 mM Tris buffer and 0.1% wt/vol EDTA; 0.5% SDS; 50 U/mL DNase and 1 U/mL RNase Protocol III: protease inhibitors (aprotinin, 10 KIU/mL, 0.1% *w*/*v* EDTA); 10 mM Tris-HCl, 0.1% SDS Protocol IV: 1% SDS; 1% Triton-X100 Protocol IV: 1% SDS and 1% Triton-X100 together	Human cardiac primitive cells	350μm-thick sections were cut from the human myocardium and decellularized with five protocols. The best result was detected from the one by 1% SDS and 1% Triton for 24 h. Reseeded dECM with human cardiac primitive cells has supported the differentiation of seeded cells toward CMs and SMCs, as explained by distinct gene expression for CMs (MEF2C, ACTC1) and SMCs (GATA6, ACTA2).

## Data Availability

Not applicable.

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
