# Peer review of "Whole-Heart Tissue Engineering and Cardiac Patches: Challenges and Promises"

_bioengineering, 2023, doi:10.3390/bioengineering10010106_

Round 1

Reviewer 1 Report

Dear Authors, 

I read with interest but also quite some difficulties your review on "Whole heart tissue engineering and cardiac patches : Challenges and promises."

Though you refer appropriately to an exhaustive amount of references, the information provided in your manuscript remained mostly descriptive and the reader will find it quite difficult to go back and forth from your table to the text in order to teese out the most informative details and facts.

The patches "story" remains a bail out strategy which comes short of the final goal : Tissue and organ regeneration.

Aside, there are quite some spelling errors (e.g. infection instead of infarction....) 

This manuscript would benefit of more deep-insighting, and focus-narrowing to gain in interest.

Author Response

Reviewer#1:

a- Though you refer appropriately to an exhaustive amount of references, the information provided in your manuscript remained mostly descriptive and the reader will find it quite difficult to go back and forth from your table to the text in order to teese out the most informative details and facts

  • Response:

Thank you very much for your comment. We have added all manuscripts from tables to the text in the revised document.

b-The patches “story” remains a bail out strategy which comes short of the final goal : Tissue and organ regeneration

  • Response:

Thank you very much for your valuable comment. We have expanded the clinical need and future directions in the revised version.

c- Aside, there are quite some spelling errors (e.g. infection instead of infarction....) 

  • Response:

Thank you very much for your attention. Please see the revised version. We have completely checked the document and fixed all spelling issues.

Reviewer 2 Report

The paper is an up-to-date review of tissue engineering and regenerative medicine in the setting of cardiovascular disease,maybe it is a revised version of a thesis or a meeting lecture.  Major approaches are described and available evidence is clearly summarized. The manuscript is well written but the real weakness is its inherent complexity that might limit readership interest,it is completely understandable by an highly specialized readership. All in all, suitable in the present form, with a concern on fruibility.  Authors might partially overcome this feature (better than a limit) adding a more detailed clinical bottom line of such complex data from regenerative medicine experiments.

Author Response

Reviewer#2: 

a-The manuscript is well written but the real weakness is its inherent complexity that might limit readership interest

  • Response:

Thank you very much for your comment. We have added new data to the document and changed the structure partially. We hope that this would make our manuscript more understandable.

b-  Suitable in the present form, with a concern on fruibility.  Authors might partially overcome this feature (better than a limit) adding a more detailed clinical bottom line of such complex data from regenerative medicine experiments.

  • Response:
  • Thank you very much for your comment. We have added the section “the role of cardiac tissue engineering in clinic” to the end of our manuscript.

Reviewer 3 Report

The review is quite general, has the advantage to provide a complete overview of all the aspects involved in cardiac tissue regeneration. 

The numerous Tables are useful, even if space consuming as they are proposed. Maybe could be reorganized.

Author Response

Reviewer#3: 

The numerous Tables are useful, even if space consuming as they are proposed. Maybe could be reorganized

  • Response:

Thank you very much for your comment. We have edited the tables and their related text in the manuscript.